# Fabrication and Evaluation of Filtration Membranes from Industrial Polymer Waste

**DOI:** 10.3390/membranes13040445

**Published:** 2023-04-19

**Authors:** Saleheen Bano, Mukesh Pednekar, Saranya Rameshkumar, Dipu Borah, Michael A. Morris, Ramesh Babu Padamati, Niamh Cronly

**Affiliations:** 1School of Chemistry, CRANN, Trinity College Dublin, D02 PN40 Dublin, Ireland; 2AMBER, SFI Research Centre for Advanced Materials and BioEngineering Research, D02 PN40 Dublin, Ireland; 3School of Physics, CRANN, Trinity College Dublin, D02 PN40 Dublin, Ireland; 4Dairy Processing Technology Centre (DPTC), University of Limerick, V94 T9PX Limerick, Ireland

**Keywords:** PVDF gels, polymer, membranes, phase-inversion, filtration, recyclability

## Abstract

Polyvinylidene fluoride (PVDF) polymers are known for their diverse range of industrial applications and are considered important raw materials for membrane manufacturing. In view of circularity and resource efficiency, the present work mainly deals with the reusability of waste polymer ‘gels’ produced during the manufacturing of PVDF membranes. Herein, solidified PVDF gels were first prepared from polymer solutions as model waste gels, which were then subsequently used to prepare membranes via the phase inversion process. The structural analysis of fabricated membranes confirmed the retention of molecular integrity even after reprocessing, whereas the morphological analysis showed a symmetric bi-continuous porous structure. The filtration performance of membranes fabricated from waste gels was studied in a crossflow assembly. The results demonstrate the feasibility of gel-derived membranes as potential microfiltration membranes exhibiting a pure water flux of 478 LMH with a mean pore size of ~0.2 µm. To further evaluate industrial applicability, the performance of the membranes was tested in the clarification of industrial wastewater, and the membranes showed good recyclability with about 52% flux recovery. The performance of gel-derived membranes thus demonstrates the recycling of waste polymer gels for improving the sustainability of membrane fabrication processes.

## 1. Introduction

Polyvinylidene fluoride (PVDF) is a semi-crystalline polymer with excellent mechanical strength, high thermal stability, good chemical and ageing resistance, and ease of processability. Owing to its outstanding properties, it has received a great deal of attention and is being used in different industrial applications [1,2]. It is one of the most widely used polymers in membrane production, and many studies have reported the development and applications of PVDF membranes, which have been extensively utilized in ultrafiltration, microfiltration and membrane distillation processes [3,4,5]. Despite having excellent properties, PVDF polymers have some limitations in membrane manufacturing, especially in the phase inversion process. In a phase inversion process, a high-concentration polymer solution closer to the ‘gelation’ phase is converted into a solid membrane in a controlled manner. Any variation in the process parameters, such as temperature, concentration, etc., can lead to an extensive formation of gels, causing these solutions to become unsuitable for membrane production. Although the phenomenon of gel formation in polymers is advantageous in some applications, an aspect that is usually overlooked is the potential detrimental effect on the structure and properties of industrial standard membranes, for which no report has been addressed to date.

Membrane technology has grown rapidly over the past several decades, expanding from water treatment [6,7] to gas purification [8], energy-related applications [9], food processing [10] and others [11,12]. Among different processes, non-solvent-induced phase separation (NIPS) is the most versatile and widely used technique for the fabrication of industrial standard PVDF membranes because it enables the production of high-performance membranes with a wide range of different characteristics [13]. Typical steps involved in membrane production applications include polymer dissolution, membrane casting, and phase inversion, for which solvents play a major role in determining the processability, morphology and performance of the final products [14,15].

One of the crucial steps during membrane fabrication via NIPS method is preparing a homogeneous solution for membrane casting. This requires strong or good solvents to dissolve polymers [16]. To produce PVDF-based membranes, polar aprotic solvents such as DMF, DMAc and NMP are the most commonly used due to their high solvent power in dissolving the polymer [5,17]. However, because of their toxic nature and environmental concerns, these solvents are expected to be prohibited for large-scale production [16,17,18,19].

According to previous reports, acetone is listed under the green solvents and is being used in different industrial scale applications [20,21,22]. In recent years, it has been found to be a suitable solvent for producing PVDF-based commercial membranes due to its low toxicity, low boiling point and ability to dissolve polymers [23,24]. In addition to low toxicity, using acetone can eliminate the issues related to high-boiling-point solvents such as their incomplete removal during the process, which may have an adverse effect on the membrane structure and industrial applications. Additionally, it is relatively less challenging to recover acetone for reuse and recycling [23].

PVDF is a semi-crystalline material and exists in different polymorphs, as defined by the stereo-regularity of the polymeric chains [24]. It has a tendency to undergo gelation depending on conditions such as the type of solvent used, temperature and time of casting during the membrane manufacturing process. A gel is a three-dimensional network of polymer chains formed by sol–gel interconversion. This interconversion can be reversible or irreversible depending on the type of interactions between the polymer chains [25,26]. It has been reported that gel formation in PVDF occurs at a faster rate when aliphatic ketones such as acetone are used as solvents [27]. Acetone has less solvation power for PVDF; hence, the weaker solvent–polymer interactions allow the aggregation of polymer chains, resulting in nucleation centers that enable other polymer chains to interact and form a network structure, entrapping the solvent [28,29]. Such polymer networks formed through physical interactions are defined as thermo-reversible gels [27].

In recent years, PVDF-based thermo-reversible gels have been considered as favorable materials for gel polymer electrolytes in Li-ion batteries due to their excellent properties [30,31]. Even while polymeric gels are gaining interest for several other applications, the other side of the coin is often ignored. As stated in previous reports [1,32,33], the gel formation in casting solution during membrane fabrication has a great impact, altering the structure and properties of the resulting membrane. To avoid the structural changes in the resulting PVDF membrane, which do not conform to the industrial standard, significant considerations are typically restricted to filtering the polymer solution to prevent faulty structures brought about by the presence of large inclusions of gels into the membranes. These filtered-out gels, produced as industrial waste by membrane manufacturing plants, are generally unused and discarded as waste. These gel materials currently end up in either landfill or incineration and add to the current level of plastic pollution. The effective utilization of these waste PVDF gels, and the addition of value, will reduce the impact on solid waste, minimize the use of virgin resources and improve the material’s circularity in general. The current study examines the reusability of PVDF gel wastes in high-value PVDF membranes suitable for filtration applications while considering the sustainability of the membrane manufacturing process. To the best of our knowledge, the recycling of gel wastes from the membrane manufacturing industry has not yet been reported.

## 2. Materials and Methods

### 2.1. Materials

Kynar K460 PVDF resin from ARKEMA GmbH (Dusseldorf, Germany) was used as the polymer. Acetone (purity 99.8%, HPLC grade) was purchased from Acros Chemicals, Ireland, and bovine serum albumin was procured from Sigma Aldrich, Wicklow, Ireland. All chemicals were used without further purification. The dairy processing wastewater sample used for this membrane performance study was collected from the Ireland-based dairy industry. This dairy wastewater sample was generated during the demineralization of whey.

### 2.2. Production of Gels in PVDF Solutions

The gels were induced in the PVDF/acetone solutions by prolonged holding at room temperature. The solid gels that were produced were further collected, dried and ground in a mill to make a fine powder of PVDF gels in order to study their further reusability (Figure 1).

In a typical process, the requisite amounts of PVDF resin and gel powders are taken separately in reaction vessels to prepare solutions in acetone with the optimum concentration of 15 wt.% [34]. PVDF solutions were prepared at 55 °C ± 2 °C by mixing under vigorous stirring for 3 h to ensure proper mixing, yielding a transparent solution without any solid content, as reported previously but with slight modification [35,36]. The selection of mixing parameters was made carefully, which might have affected the membrane morphology [33,37]. Following dissolution, the solutions were cooled to 25 °C ± 2 °C and kept for 1 h before the membrane fabrication process.

### 2.3. Fabrication of PVDF (Resin and Gel) Membranes

The membranes were prepared by the immersion precipitation technique, as mentioned previously but with slight modifications [33,38]. In order to fabricate membranes, the prepared polymer solutions were cast onto a glass plate using casting knife. The cast membranes were then instantly immersed in a coagulation bath containing acetone and water mixture as non-solvent, with an optimum weight ratio of 60:40. The bath temperature was maintained at 25 °C. The fabricated membranes were further kept in deionized water overnight to ensure the complete release of solvent. Finally, the obtained membranes, with a thickness of ~80 µm, were dried at room temperature overnight. The dried membranes were stored in airtight bags for further use. In the present work, gel membranes are denoted as GM, whereas membranes made of PVDF resin are marked as RM.

### 2.4. Characterization Methods

#### 2.4.1. ATR—Fourier Transform Infrared (FTIR) Spectroscopy

To manifest the chemical structures of samples, FTIR spectra were recorded in the range of 500–4000 cm^−1^ at a resolution of 4 cm^−1^ and with 16 scans, using a Spectrum 100 Perkin Elmer spectrophotometer (Waltham, MA, USA) in ATR mode.

#### 2.4.2. Wide-Angle X-ray Diffraction (WAXD)

A Bruker Panalytical (X′pert^3^, Malvern, UK) X-ray diffractometer with a CuKα (λ = 1.5406 Å) radiation source was used to record the X-ray diffraction (XRD) patterns of samples over an angular range of 5–40° at a scan rate of 0.025° min^−1^. The diffractogram identifies the presence of crystalline phases in the samples and the percent crystallinity can be estimated as per Equation (1), as follows:(1)$$\%\;Crystallinity=\;\frac{A_{c}}{A_{c}+A_{m}}\times 10$$
where *A_c_* and *A_m_* represent areas under crystalline and amorphous regions, respectively. The described areas were obtained after deconvoluting the diffractogram specific to each sample.

#### 2.4.3. Viscosity Test

To understand the solution behavior, the viscosity of each casting solution was determined by means of Brookfield viscometer (Model DV2T from Middleborough, MA, USA) at room temperature (25 °C) and at a speed of 10 rpm using a LV63 spindle. The average viscosity was calculated in a two-fold determination for each sample.

#### 2.4.4. Rheological Test (Shear Viscosity)

In the present context of understanding the reusability of gels, the shear viscosity of gel and resin solutions was measured at room temperature using a rheometer (Model AR2000, TA Instruments, New Castle, DE, USA) with a concentric cylinder and solvent trap to avoid any solvent loss due to evaporation

#### 2.4.5. Light Scattering Analysis: Dynamic Light Scattering (DLS)

DLS analysis was performed to reveal the impact of gels in terms of their reusability by determining the size of the micro-gels present in each of the casting solutions of PVDF resin and gel powders. This study was carried out on 1% solution using a zetasizer NanoZS Malvern instrument (Malvern, UK) at a wavelength of 639 Å.

#### 2.4.6. Turbidity Test

The solution property of PVDF resin, as well as gel powder, was further analyzed by measuring the turbidity of each of the casting solutions using a Lovibond TB 100 tintometer from Amesbury, UK. The turbidity measurement was performed at room temperature with respect to time until the solution became turbid and densely cloudy in appearance. This study is an effective tool to understand the stability behavior and unfold the phenomenon of further gelation in a PVDF gel solution.

#### 2.4.7. Molecular Weight Analysis

To understand the solution behavior of PVDF samples, molecular weight properties were analyzed by means of gel permeation chromatography using a GPC Viscotek TDA 305 instrument (Malvern, UK). For analysis, samples were prepared at a concentration of about 2 mg mL^−1^ by dissolving them in dimethyl sulfoxide (DMSO) conditioned with 0.1 *w*/*v*% of lithium bromide (LiBr). The solutions were further filtered with 0.45 μm PTFE syringe filters. All samples were detected at standard conditions of eluent flow rate (0.7 mg mL^−1^), column temperature (60 °C) and injection volume (50 µL).

#### 2.4.8. Differential Scanning Calorimetry (DSC) Analysis

The Perkin Elmer DSC 200 F3 Differential Scanning Calorimeter (Waltham, MA, USA) was used to conduct DSC measurements in order to examine the melting and crystallization behavior of PVDF samples. The thermogram was scanned by heating about 4 mg of samples between temperatures of 50 °C and 250 °C at a rate of 10 °C/min in a nitrogen atmosphere with a gas flow rate of 19.8 mL min^−^^1^. Based on the melting enthalpy of ideal PVDF crystals (ΔH_f_ = 104.7 J g^−^^1^), crystallinity was evaluated [39].

#### 2.4.9. Field Emission Scanning Electron Microscopy (FE-SEM)

A Zeiss Ultra plus (FEG Quanta 6700) Field Emission Scanning Electron Microscope (Cambridge, UK) was used at an accelerating voltage of 5 kV and a working distance of 6–7 mm to analyze the surface and cross-sectional morphology of membrane samples. For cross-section imaging, the membranes were fractured in liquid nitrogen. Before analysis, the samples were sputtered with gold using a vacuum sputter coater. The pore size diameter was determined using ImageJ software IJ 1.46r developed by National Institute of Health (Bethesda, MD, USA).

#### 2.4.10. Pore Size Determination via Mercury Porosimetry

The Autoscan-33 Porosimeter (Quantachrome, Hook, UK) was used to measure the membrane sample’s macro-porosity using mercury porosimetry, with a default contact angle of 140°. The tests involved the infiltration and extrusion of mercury under pressures ranging from 0 to 33,000 PSI in order to evaluate the pore size and pore distribution.

#### 2.4.11. Atomic Force Microscopy (AFM)

AFM was used to characterize the surface roughness of the samples. The topography of membrane samples was scanned in non-contact tapping mode, using an AC160TS silicone cantilever type, on a Park XE-100 (Park Systems, Suwon-si, Republic of Korea).

#### 2.4.12. Porosity Measurement

The overall porosity of the membranes was determined as per Equation (2) [13]. (2)$$\mathrm{Porosity}\;\left(\%\right)=(1-\rho_{m}/\rho_{PVDF})\times 100$$
where *ρ_m_* is the density of membrane, determined gravimetrically by weighing a sample of known area and thickness.

#### 2.4.13. Contact Angle (CA) Measurement

The membrane’s hydrophobicity was examined by dynamic CA measurements (a custom-built system) at three different regions of each sample. The images of water CAs were captured using high-speed camera with a 60 Hz sampling rate. The water was dispensed through a gauge needle with 135 μm OD at a flow rate of 5 nL/s with a droplet volume of 50–100 nL.

#### 2.4.14. Membrane Performance Characterization

The performance of in-house-developed PVDF resin and gel membranes was evaluated in a crossflow filtration system. As shown in Figure 2, the benchtop crossflow filtration unit includes SEPA CF cell assembly (with membranes) connected to a feed tank, feed pump and pressure gauges. The SEPA CF assembly includes a stainless steel cell, anodized cell holder and hydraulic hand pump.

The flow diagram of crossflow filtration is shown in Figure 3, where feed is passed through the membrane in a tangential direction to the membrane surface at a desired feed flow rate and inlet pressure, and the filtrate passing through the membrane, i.e., the permeate, is collected separately in a permeate collector, while the feed fraction retained on the membrane surface, i.e., the retentate, is returned to a feed tank to further continue the filtration.

In this study, the pre-cut membranes of size (140 × 190 mm) with an active separation area of 140 cm^2^ were used for their performance evaluation. In each trial, before actual analysis, the membranes were first equilibrated with the feed conditions by recirculating the permeate to the feed tank for 20 min at a constant pressure of 3 bar. After completing the membrane equilibrium, the permeate was collected separately and flux was measured at certain time intervals to assess the membrane’s performance in terms of flux drop. During the membrane filtration, the feed temperature was maintained at a constant by connecting the feed tank to a temperature-controlled water circulator.

Filtration studies using BSA solution (1 gL^−1^) as a feed were performed on the above-mentioned filtration set-up by employing the fabricated membranes. Initially, deionized water was passed through the membranes to determine and compare their pure water fluxes. Each membrane filtration analysis was performed for 2 h at a constant temperature of 25 °C and a transmembrane pressure of 3 bar with a feed flow rate of 3.2 Lmin^−1.^ The permeate was collected every 10 min, and at least three readings were taken to determine the change in flux over the period of filtration.

The permeate flux through the membranes was determined using Equation (3).
(3)$$\mathrm{Flux}\;\left(J\right)\;\left(Lm^{-2}h^{-1}\right)=V/\left(A\times t\right)$$
where *V* is the volume of feed in a liter (L), *A* is the effective area of membranes (m^2^) and *t* is the filtration time (h).

The rejection efficiency of the membranes was quantified based on BSA concentration in the permeate and was calculated using Equation (4) to investigate the membrane selectivity.
(4)$$\mathrm{Rejection}\;\left(\%\right)=(1-C_{p}/C_{f})\times 100$$
where *C_f_* and *C_p_* are the BSA concentrations in feed and permeate. The permeate concentration was determined by estimating the absorbance of BSA at 285 nm in UV–visible spectroscopy using a Perkin-Elmer lambda 1050 spectrophotometer (Dublin, Ireland).

After the completion of the filtration study, the membrane surfaces were first cleaned with deionized (DI) water for 1 h to remove the loosely bound pollutant layer and then chemically treated with 0.1 M sodium hydroxide solution to remove the strongly adsorbed pollutant from membrane surfaces and pore walls. The alkali solution was recirculated for 1 h at 40 °C and then the membrane was further washed with DI water for 2 h until pH neutrality. The pure water flux (*J*_2_) of the cleaned membranes was measured again to evaluate the flux recovery ratio (*FRR*) using Equation (5).
(5)$$\;FRR=\left(J_{2}/J\right)\times 100$$

The performance, durability and reusability of the gel membrane was further estimated using industrial dairy processing wastewater (pH 2.6–3). The membrane was tested on the same crossflow filtration unit and under the same conditions as mentioned above. The analysis process included: (1) the measurement of the pure water flux of fresh membrane; (2) membrane treatment with wastewater and flux measurement; (3) membrane cleaning by physical and chemical treatment, followed by pure water flux measurement; and (4) a second cycle of analysis on the gel membrane, run in the same way, to demonstrate its reusability. Turbidity tests and a comparison with RM for the first cycle of operation were also used to assess the GM membrane’s capacity for filtration.

## 3. Results and Discussion

### 3.1. Analysis of Processed Dried Gels

#### 3.1.1. Structural Analysis of Gels

As illustrated in the FTIR spectra (Figure 4a), both PVDF resin and gels exhibit similar spectral bands. The presence of bands around 764 cm^−1^ and 1404 cm^−1^ is characteristic of absorption bands assigned to the α-crystalline phase of PVDF [40]. This suggests that there is no significant variation between the molecular phases of the two and the chemical structure of the gel remains the same as that of PVDF resin after processing.

The findings of the FTIR study were supported by XRD analysis, as the obtained diffraction patterns (Figure 4b) showed the presence of three prominent peaks at 2θ = 18.3°, 19.8° and 26.5° corresponding to the (020), (110) and (021) planes, respectively, of the α-crystallite phase of PVDF [41]. The X-ray studies further confirmed that the gel powders, obtained after processing, predominantly exist in the alpha phase, similar to the PVDF resins.

The percent crystallinity calculated also falls in the same range as for the PVDF resin, as summarized in Table 1. However, the crystallinity of the processed gel was moderately higher than that of the PVDF resin, which can be explained by the swollen nature of the gel, which may allow crystallite refinement through increased chain motion in the more amorphous regions of the gel [28].

As shown in the DSC scans (Appendix A) obtained from the thermal study, the melting enthalpy and melting temperature of gel crystallites were found to be comparable with the PVDF resin, providing more evidence to validate the findings of the structural analysis. Table 2 lists the percentage of crystallinity, derived from melting enthalpy, which follows a similar trend as obtained by analyzing the X-ray diffraction patterns of the PVDF resin and gels.

#### 3.1.2. Solution Properties

The dried gels were found to have solution behavior similar to that of PVDF resins as revealed from viscosity and turbidity analysis. The obtained average viscosity values for both the solutions of PVDF resins and gel powders were comparable to one another, as summarized in Table 1. However, the solutions of gel powders had a small increase in viscosity compared to that of PVDF resins. This can be attributed to the slightly higher molecular weight and crystallinity of the gel powders.

The trend observed for absolute viscosity was further supported by the similar trend in shear viscosity from the rheological analysis of the solutions of gel and PVDF resin under the same concentration and temperature conditions. The shear viscosity for both the solutions falls in the same range and tends to decrease with an increase in the shear rate at a constant temperature of 25 °C (Appendix A).

Turbidity measured in NTU (nephelometric turbidity unit) helps to predict the phenomenon of gel formation over time at constant temperature and hence the stability of PVDF solutions. The turbidity value or cloudiness tends to increase with time, following a similar trend for both the PVDF solutions, revealing that gel powders have similar solution stability or behavior as obtained for solutions of PVDF resins under the same conditions (Figure 5). However, the turbidity of the gel solution was slightly higher than that of PVDF resins due to its molecular properties, as mentioned in the previous section. The onset of gelation occurred after 4 h of storage, suggesting the good stability of both solutions and that a longer duration of holding enhances the level of gelation. It can be presumed that with an increase in time, the tendency of polymeric chains to aggregate increases, resulting in a further increase in the cloudiness or turbidity value. An obvious difference in cloudiness was observed after 8 h of storage and the solutions became solidified after holding for more than 12 h.

The gel solutions possessed slightly more turbidity, indicating more sensitivity towards gelation, which can be attributed to the relatively higher crystallinity and molecular weight, facilitating more nucleation sites by enhancing the molecular networking responsible for the increase in cloudiness.

PVDF is a crystalline polymer and is prone to gel formation through polymer chain interactions; thus, the presence of gels at the molecular level in PVDF solutions cannot be avoided. The DLS measurement was further performed on the dilute solutions of both resin and gel powders to compare the size of micro-gels in both solutions at a given temperature. As portrayed in Figure 6, both the gel and resin solutions have similar micro-gel size distributions. However, the comparatively higher intensity for gel powders further supports the turbidity data, predicting their slightly higher sensitivity towards gelation.

### 3.2. Analysis of Fabricated Gel Membrane

#### 3.2.1. Physical Properties

The structural analysis shows that both the resin and gel membranes exist in the same α-crystalline phase, as revealed from the FTIR spectrum and X-ray diffractogram (Appendix A), since the membrane was fabricated from the PVDF solutions in acetone, and it has been reported that acetone predominately induces the α-phase in fabricated PVDF membranes [42]. This shows that the molecular integrity of the membrane is the same as that of the original substance.

Moreover, both the resin and gel membranes possess similar thermal properties and are comparable to the initial materials (Appendix A and Table 2), upholding the retention of molecular structure in the obtained products. The melting enthalpy and crystallinity obtained for the membranes are relatively higher than those of the initial materials, which can be related to the molecular orientation and packing of polymer chains induced during the phase inversion of the membrane fabrication process. This leads to a more ordered and compact crystalline structure, resulting in a higher melting enthalpy and crystallinity of the resulting membranes [33,43,44].

#### 3.2.2. Morphological Properties

Figure 7 depicts the morphology of the top surface and the cross-section of prepared membranes. The morphological analysis revealed that both membranes possess similar porous morphology with interconnected pores. However, the membranes fabricated from the gel powders showed the presence of slightly denser polymer networking.

The most typical feature of the fabricated membranes is the bi-continuous structure with the complete absence of finger-like pores or macro-voids. The presence of interconnected porous morphology in the fabricated membranes has also been reported previously, which signifies good mechanical strength in the membranes [33,43,44]. The cross-sectional view reveals that throughout the thickness (from top to bottom), the morphology is symmetrical due to the absence of a dense skin layer, which could be the effect of using a soft non-solvent to induce polymer precipitation during phase inversion for membrane formation [34,45]. It has been explained previously that the selection of a weak non-solvent may reduce the rate of liquid–liquid demixing during the phase inversion [3,37,45], which may result in the absence or bare appearance of the skin layer, causing the symmetrical morphology of membranes.

The formation of a bi-continuous structure with interconnected pores is an important parameter to represent the potential of fabricated GM membranes to be used in the microfiltration (MF) process, as described in previous works [46]. According to Liu et al. [44], this bi-continuous morphology is frequently seen in the commercially available PVDF-based MF membranes.

The statistical data for surface morphology suggest that the resin membranes possess a slightly wider range of pore size distribution than the gel membranes; both of the membranes possessed a maximum distribution that fell in the range of 200 nm. The mean pore size diameter for the resin membrane (280 ± 177 nm) was remarkably higher than that of gel membranes (270 ± 155 nm). The higher solution viscosity of the gel powders can be used to corroborate the observed reduction in pore size in the SEM image, as well as in the porosity measurements (Table 3). This is consistent with the previous reports [47,48], which describe that diffusion between the solvent and non-solvent at the nascent membrane surface might be hindered during the phase inversion in the case of more viscous solutions, resulting in a relatively less porous structure.

The difference in the pore size of the two membranes can further be ascribed to the nucleation density present in the doped solutions, as reported previously [33]. It can be presumed that processed gels contain a large population of pre-nucleation aggregates in the casting solution, which further nucleate and grew when they encounter the non-solvent. The crystallinity data, as summarized in the previous section, support the presence of a relatively greater nucleation density than in the resin. The higher density of nucleation sites might enhance the rate of crystallization or gelation phenomenon more than the liquid–liquid demixing during the phase inversion process using a non-solvent, which can be ascribed to the relatively denser morphology with a smaller pore size for gel membranes [49].

The topography of the membrane’s top surfaces was also examined by atomic force microscopy, as shown in Figure 8. RM displayed more roughness (Ra = 50.3 nm, Rq = 67.8 nm) than GM (Ra = 48.1 nm, Rq = 60.9 nm), which is in line with SEM findings. The surface roughness is defined by the vertical distance that the scanner moves, and when a surface has more pores or nodules, the tip moves across a wider range, increasing the roughness parameters, as in the case of RM.

However, during the fabrication process, a significant change in physical properties at the molecular level was not observed, as revealed from structural and thermal analysis. This can be used for micro-filtration applications, as discussed in a later section.

#### 3.2.3. Pore Size Analysis by Mercury Porosimetry

The data obtained from the morphological analysis were further supported by characterizing the pore size of the membranes via porosimetry. Depending on the pore diameter, materials can be classified as macro-porous with >50 nm pore diameter, meso-porous (2–50 nm) or micro-porous (<2 nm) [50]. The fabricated membranes investigated in the present context fall in the category of macro-porous materials, as depicted in Figure 7. The sample’s macro-porous morphology is generally comparable, with pore diameters ranging from 200 to 300 nm, as shown in the SEM micrograph (Figure 7c,g).

Mercury porosimetry analysis was carried out to determine the macro-porosity of membranes. Figure 9a,b depict the mercury infiltration and the consequent pore size distributions, respectively, for resin and gel membranes.

It is observed that for RM, mercury intrusion starts at ~5 μm and a steep intrusion commences at ~0.7 μm (700 nm), which continues up to 0.02 μm (20 nm) (Figure 9a), indicating that the majority of pores fall between 0.020 and 0.7 μm in size. The curve levels off after 0.02 μm and no further mercury intrusion was seen. For GM, the bulk of the pores are found in the range of 0.016–0.6 μm, as evidenced by the sharp mercury infiltration that occurs around 0.6 μm (600 nm) and lasts until 0.016 μm (16 nm). The steep intrusion profile indicates open, interconnected porosity throughout the membrane and is compatible with the rapid transport of mercury through a sample.

A unimodal pore size distribution with pore sizes ranging from 0.02 to 0.6 μm is seen in Figure 9b for RM. The pore size distribution for GM is similar to that of RM, encompassing pores of 0.018–0.7 μm. For the resin membrane, the mean flow pore size was found to be 230 nm, which is comparatively more than the gel membrane, with a mean pore size of 195 nm. Overall, the results of mercury porosimetry are consistent with the findings of morphological analysis by SEM.

#### 3.2.4. Filtration Properties

The performance of the fabricated membranes, in relation to flux permeability, selectivity and flux recovery efficiency, was also evaluated in order to reveal more specific properties of membranes in relation to the treatment capacity, as shown in Figure 10 and Table 3.

The flux permeability of GM follows a similar trend to that of RM (Figure 10); however, the pure water flux of RM was comparatively more than GM, as shown in Table 3. In general, water flux is determined by the pore size, porosity and hydrophilicity of the membranes [51,52]. As both membranes in the present study are hydrophobic, the observed difference in the permeability of the membranes can be correlated with the variation in the porous structure, revealed from the morphological analysis. Therefore, the relatively denser morphology with a smaller pore size of the GM may be accountable for its lower water flux and comparatively higher selectivity than RM, which is in good agreement with previous reports [41,51].

The relatively higher porosity and larger pore size for RM might assist water molecules in passing through the membrane and accordingly increase the permeability; however, a more porous structure had a negative impact on the retention of the solute of interest, and prior research supports this idea [52]. The mean pore size and pure water flux, as well as the BSA retention data, demonstrated the suitability of the gel membrane for the filtration process when compared with previously reported works on PVDF-based membranes, as listed in Appendix A. However, the further modification of gel membranes based on structure optimization is necessary for practical application.

Membranes were also investigated for the effectiveness of fouling resistance with respect to BSA. The membrane’s surface hydrophilicity is crucial to its ability to resist protein fouling during filtration; however, because no hydrophilic additives were introduced in the present study, the fabricated PVDF-based membranes were essentially hydrophobic, as demonstrated by the contact angle measurements. The contact angle of GM was 105°, which is lower than the angle of 110° of RM. The higher contact angle of RM can be related to its surface roughness, as seen from AFM (Figure 8). The BSA permeation flux for RM and GM declined by 36.9% and 41.8%, respectively, of the initial value after 1 h, as shown in Figure 10b. The dramatically higher decline in BSA flux may be related to the high hydrophobicity and lower porosity of the GM membranes, which may lead to the rapid deposition of foulant layer on the membrane surface. This would reduce the surface openings, resulting in less porosity and smaller pores, and thus reducing the membrane permeability.

Meanwhile, the flux recovery ratio of the gel membrane was marginally higher but still equivalent to that of the resin membrane, predicting good fouling resistance. This may be correlated with the roughness variation in the membrane surface. The antifouling performance tends to decrease with an increase in roughness, as the contaminant may accumulate in the valleys of the rough membrane surface [53]. The surface roughness of the gel membrane is slightly lower than that of the resin membranes due to enhanced inter-chain interactions which suppress the height variations. As per previous reports, in cases of membranes with a smoother surface, the efficient filtration area is reduced and a higher interaction energy barrier between foulants and membrane surface is anticipated, which would prevent fouling. This also makes it easier for protein molecules to dislodge, thus extending the operational lifetime of membranes [48,53]. However, further work is needed in the future to validate this claim.

### 3.3. Wastewater Fouling and Recyclability of Gel Membranes

To further evaluate the feasibility of the in-house-prepared gel membranes, the filtration experiments were carried out using dairy wastewater on the same set up at the same operating conditions for two hours. The results in terms of relative flux permeability with respect to time for wastewater are depicted in Figure 11a. The permeation flux for GM declined by 56% of the initial value, compared with 37.5% in the case of RM after 2 h of the experiment. The decline in permeability can be attributed to membrane pore size and hydrophobicity. The smaller pore size and high hydrophobicity may lead to an increase in intrinsic membrane resistance and might allow for more pollutant molecules to accumulate on the membrane surface, resulting in higher fouling and thus affecting the membrane flux and selectivity.

Additionally, the permeate samples collected at certain time intervals were also analyzed for the turbidity measurements. When compared with the turbidity of the feed solution (83.76 NTU), the initial reduction in turbidity following filtration with GM was 92%, as opposed to 68% for RM, demonstrating the good filtration performance of GM over RM, which can be attributed to the difference in the membrane morphologies. The small pore size of GM might improve the retention of pollutants on the membrane surface, leading to a higher reduction in permeate turbidity and the improved clarification of wastewater.

As represented in Figure 11b, the turbidity of the permeate stream tends to decline faster for the gel membrane with respect to time and to become constant after 1 h. Again, the observed trend in turbidity can be ascribed to the relatively lower porosity and smaller surface pore size for the GM. The higher NTU value for the collected permeate at the initial stage of filtration with both membranes can be explained by the presence of smaller pollutant components that may pass through membranes under pressure. However, the development of a pollutant layer on the surface of the membrane may restrict them from passing on, resulting in a decrease in NTU values over time, which at later stages may become too low to be measured by the instrument. Figure 11b also encapsulates the difference between the two states before and after filtration. The permeate was almost transparent after filtration and distinct variation was not observed after 1 h of filtration.

The retention ratio of the membrane was also calculated in terms of the turbidity of the feed and permeate using Equation (3), as mentioned in the experimental section. A retention ratio of 99% was obtained for GM, which was relatively more than the value of 96% obtained for RM. The higher retention and lower flux for GM can again be attributed to its morphology, with relatively lower porosity and a smaller pore size, as well as to its hydrophobicity, as explained previously [54].

The membranes were cleaned as described in the experimental section, followed by the measurement of pure water flux to define the fouling resistance of the produced membranes. Interestingly, despite greater flux loss, the *FRR* for GM was 68%, which was relatively more than the value of 65% obtained for RM. This demonstrates that the GM possesses considerable antifouling capability. Moreover, the relatively higher *FRR* value may be due to the membrane surface being relatively less porous than RM and it therefore being easier to form a cake layer of loosely bound pollutants on the membrane surface that can be removed through washing. Contrarily, the pollutants may enter the pores and be deposited or absorbed on the membrane’s surface or the pore wall of RM with larger pores, which might lead to plugging and make it difficult to remove, resulting in a lower *FRR*, which is in accordance with a previous study [41,55]. However, to justify the practical applicability and antifouling property of the membrane, additional work will need to be carried out in the future to analyze the nature of fouling and membrane robustness with respect to different cleaning agents, in order to improve membrane performance for commercial application.

A second cycle of filtration using dairy wastewater was run using GM for another two hours at the same operating conditions to analyze the durability and reusability of gel-derived membranes. The flux loss obtained in cycle 2 was 72% for a fouled membrane, which was 16% more than that observed in cycle 1 for a fresh gel-derived membrane. A flux recovery of 52% was attained for the fouled gel membrane after the second cycle of operation, which was sufficient to demonstrate its viability in membrane separation technology. The SEM images of the fresh GM and the fouled membrane after the second cycle of analysis are shown in Appendix A, which represent minimal changes in morphology after the second run, supporting the longevity of gel membranes and their ability to be recycled.

## 4. Conclusions and Future Outlook

To understand the reusability of PVDF gels produced as waste at the industrial scale, gels were induced in a PVDF solution at the lab scale. The obtained gels were further analyzed and used for membrane fabrication.

The property analyses of the gels and, thus, the fabricated membranes showed no significant change in the structural, thermal or morphological properties when compared with the original PVDF resin and its membrane. The bi-continuous morphology, mean pore size and membrane performance, including a considerable water permeation flux, rejection efficiency and feed recovery ratio, demonstrated that the membranes fabricated from PVDF gels are feasible for use in membrane filtration technology. However, further study with a prime focus on the long-term performance of membranes, with respect to their fouling resistance, durability and reusability in practical applications, is currently being carried out and will be presented in the near future.

It can be concluded that the solidified gels obtained from industrial waste can be further processed for membrane fabrication and have the potential to be used in filtration applications by means of the effective control of the process parameters.

## Figures and Tables

**Figure 1 membranes-13-00445-f001:**
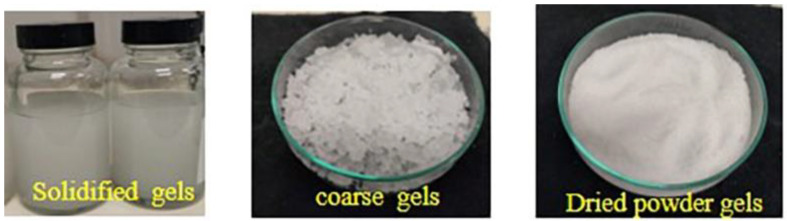
Processed fine powder of PVDF gels.

**Figure 2 membranes-13-00445-f002:**
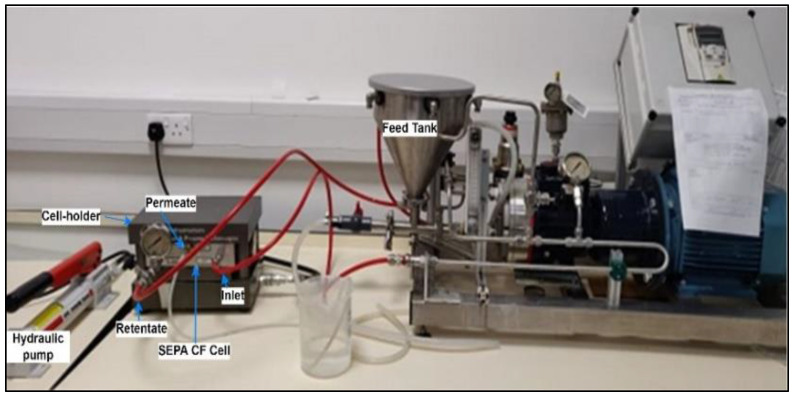
Laboratory setup of crossflow filtration.

**Figure 3 membranes-13-00445-f003:**
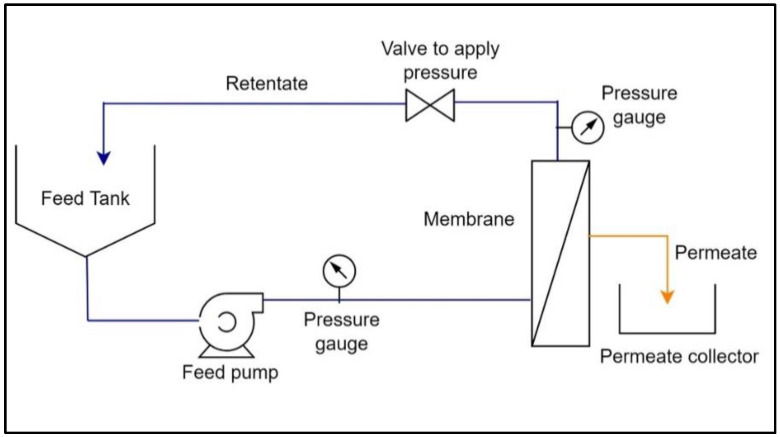
Flow diagram of the crossflow filtration.

**Figure 4 membranes-13-00445-f004:**
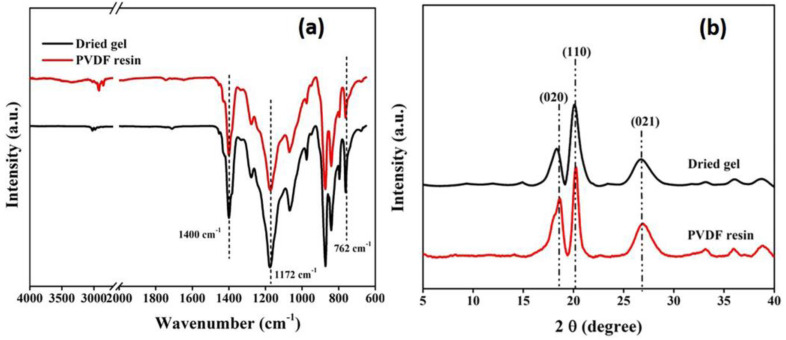
(**a**) FTIR spectra and (**b**) XRD pattern of dried gel and PVDF resin.

**Figure 5 membranes-13-00445-f005:**
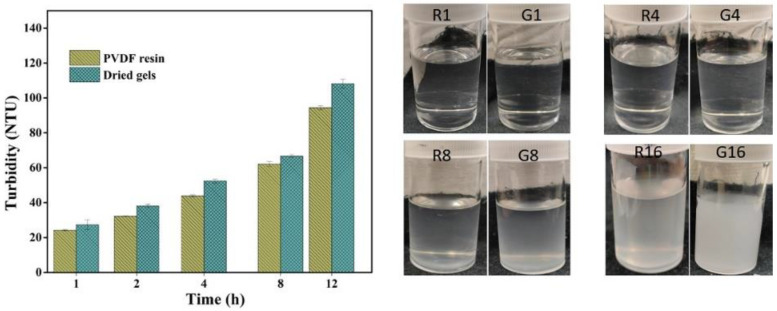
Solution stability test of resins (R) and gels (G) (the numbers represent the storage time in hours).

**Figure 6 membranes-13-00445-f006:**
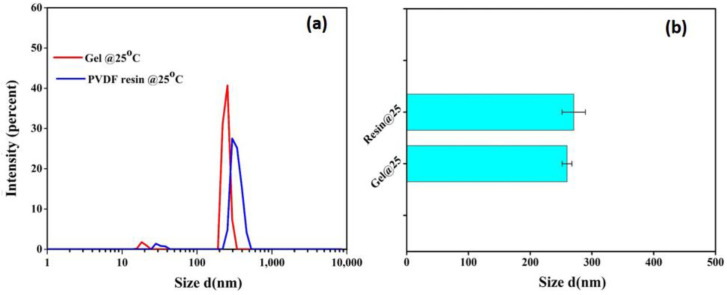
(**a**) Size distribution and (**b**) average size of microgels in dilute solution of PVDF resin and gel powders at 25 °C.

**Figure 7 membranes-13-00445-f007:**
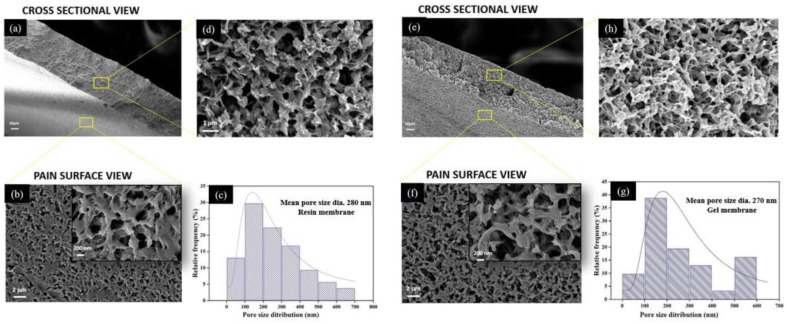
SEM micrograph at different magnifications for resin membrane ((**a**) cross-section at low magnification, (**b**) plain surface, (**c**) statistical data of pore size distribution and (**d**) cross-section from top to bottom at high magnification) and gel membrane ((**e**) cross-section at low magnification, (**f**) plain surface, (**g**) statistical data of pore size distribution and (**h**) cross-section from top to bottom at high magnification).

**Figure 8 membranes-13-00445-f008:**
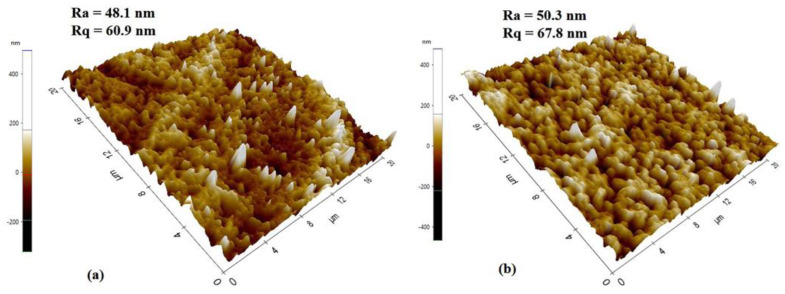
AFM scan images of (**a**) gel and (**b**) resin membranes.

**Figure 9 membranes-13-00445-f009:**
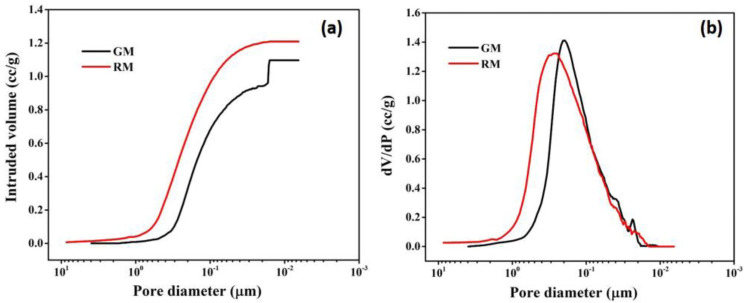
Pore size characterization: (**a**) mercury intrusion curve; (**b**) pore size distribution curve of resin and gel membrane using mercury porosimetry.

**Figure 10 membranes-13-00445-f010:**
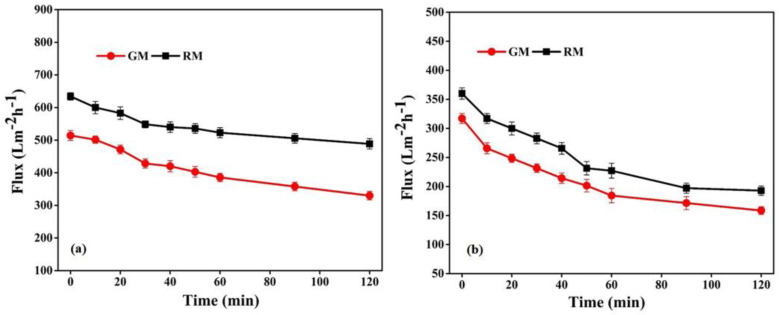
Flux with respect to (**a**) pure water and (**b**) BSA solution for resin and gel membrane.

**Figure 11 membranes-13-00445-f011:**
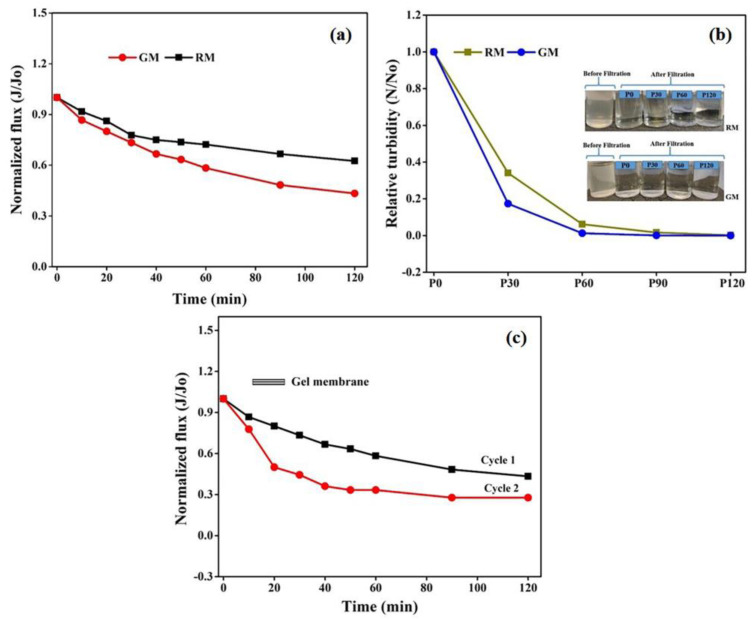
Membrane permeability (**a**) with respect to wastewater, (**b**) turbidity analyses of permeate stream over time and (**c**) relative flux of the gel membrane after two run cycles.

**Table 1 membranes-13-00445-t001:** Physical properties of the PVDF samples.

Sample	% Crystallinity	Average Molecular Weight (Mw) Daltons	Polydispersity Index (PDI)	Average Viscosity (cP) @25 °C @10 rpm
PVDF resin	33.36 ± 0.004	664,000	3.06	984 ± 1.43
Gel powder	35.02 ± 0.001	684,000	3.58	1008 ± 2.22

**Table 2 membranes-13-00445-t002:** Thermal properties of the PVDF samples and membranes.

Sample	Melting Temperature (Tm) (°C)	Melting Enthalpy (ΔHm) (J/g)	% Crystallinity
PVDF resin	159.49	33.04	31.55
Gel powder	159.83	33.21	31.72
RM	159.46	37.34	35.67
GM	158.64	37.91	36.21

**Table 3 membranes-13-00445-t003:** Membrane properties and performance analysis.

Sample	Mean Pore Size (μm)	Porosity (%)	Contact Angle (°)	Average Flux (LMH)	Retention (%)	*FRR* (%)
RM	0.23 ± 0.11	69.18 ± 4.09	110 ± 3.33	591.43 ± 15.18	25.41 ± 2.32	82.5
GM	0.19 ± 0.08	66.78 ± 6.12	105 ± 3.17	478.92 ± 12.30	26.88 ± 5.03	83.1

## Data Availability

This article does not contain new data.

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
