# Peer review of "Fabrication and Evaluation of Filtration Membranes from Industrial Polymer Waste"

_membranes, 2023, doi:10.3390/membranes13040445_

Round 1
Reviewer 1 Report
This manuscript demonstrated that gel was induced from PVDF solution and used for membrane manufacturing. The author used a variety of characterizations to show that PVDF gel had no significant structural change compared with the original PVDF resin. From the analysis of membrane filtration performance, it can be seen that the membrane prepared by PVDF gel is feasible in membrane filtration technology. However, there are still some content and format issues that need to be modified and improved:
1. This research aims at the recycling gel waste generated in the membrane production process. However, there is a problem of pollution in the filtration and collection of gel, and its composition is often more complex, making it difficult to reuse. The author directly used PVDF resin to prepare the gel, but the novelty and practicality of this method are still insufficient.
2. In the filtration experiment of dairy wastewater, the retention rate of dairy wastewater by GM is 99%, while the flux recovery rate is only 52%. It can be seen that the membrane pollution caused by the deposition of pollutants on the membrane surface or the plugging of membrane pores is difficult to be alleviated by simple chemical cleaning, which indicates that the membrane's pollution resistance is not good, and its recyclability is not strong.
3. Lines 112 and 126, how are "with optimization concentration of 15 wt.%" and "with an optimization weight ratio of 60:40" determined? Have any other relevant experiments with different proportions been done?
4. In Figure 4, it is suggested to unify the drawing color. A more detailed legend description such as (a) and (b) are suggested in Figure 6. It is recommended to unify the letter size format in the legend of the manuscript, such as (a) and (A). Please beautify the chart appropriately.
5. Line 373, the average aperture of GM is 240 nm, while it is 270 nm in Figure 8 (f). Please check the data.
6. Turbidity of dairy wastewater was reduced by 92% after filtration by GM, which is higher than RM. It is stated in the article that "......., defining good filtration performance for the gel membranes as the majority of the pollutant get filtered out after compaction.". the good filtration performance of GM is mainly due to the physical screening function of gel membrane and the formation of dirt resistance on the membrane surface, rather than because the majority of the pollutant get filtered out after compaction. This statement needs to be more rigorous.
Author Response
Please see the attachement

Reviewer 2 Report
The authors investigated the reusability of PVDF gels produced as waste at industrial scale. The polymer gels were used to prepare filtration membranes. The authors conducted extensive characterization of the prepared membranes and tested their performance in terms of flux, BSA rejection and reusability. Their findings demonstrated the recycling of waste polymer gels for improving the sustainability of membrane fabrication processes. Therefore, the topic of the manuscript can be considered as interesting for the readers. The manuscript is properly structured. Introduction highlights well the relevance of the study and specific research motivations. Applied methods are adequate and accepted in science and practice, as well. Materials and methods are clearly given. The manuscript contains significant and interesting results that have relevance not just for the science but also for the practice. Having said that, the following is suggested to improve the quality of the review:
1. The manuscript contains some writing errors in words and grammar. Please check carefully and modify them.
2. Line 41 – 44: Please name the applications where gel formation is advantages and provide references.
3. Line 113 – 115: Please highlight the modification. Experimental section needs to be detailed for reproducibility.
4. Line 159: Please provide model, supplier and country for the zetasizer used for DLS studies. This information must be provided for all instruments used in the study.
5. Line 175: The unit for volume is capital L. The must be space between values and units except between a value and %. Please correct this throughout the manuscript.
6. Line 224 – 226:At what pressure was the compaction performed?
7. Figure 6: Please add error bars for the bar graph. This needs to be done for all other measurements reported in the study.
8. Figure 7: In my opinion, this figure can be moved to supplementary information as it is similar to Figure 4.
9. Table 2: The differences between melting enthalpy of resin and powder vs RM and GM need to be discussed in more detail. Same for Crystallinity which increases for the membranes compared to the powder and resin.
10. Figure 8: Some texts in the figures cannot be read, please improve the font. Further, cross-section must show the entire membrane from top to bottom. Further, the working distance and acceleration voltage needs to be reported.
11. Table 3 must include standard deviation.
12. Fouling of RM and GM membranes was not started at same initial flux, yet fouling is controlled by flux. In my opinion the comparison for fouling behavior is not fair. Please comment on this.
13. What are the values for reversible and irreversible fouling for the RM and GM membranes?
Author Response
Please see the attchement
